# Administration of *Bifidobacterium breve* Improves the Brain Function of Aβ_1-42_-Treated Mice via the Modulation of the Gut Microbiome

**DOI:** 10.3390/nu13051602

**Published:** 2021-05-11

**Authors:** Guangsu Zhu, Jianxin Zhao, Hao Zhang, Wei Chen, Gang Wang

**Affiliations:** 1State Key Laboratory of Food Science and Technology, Jiangnan University, Wuxi 214122, China; su1994112@163.com (G.Z.); zhaojianxin@jiangnan.edu.cn (J.Z.); zhanghao@jiangnan.edu.cn (H.Z.); chenwei66@jiangnan.edu.cn (W.C.); 2School of Food Science and Technology, Jiangnan University, Wuxi 214122, China; 3International Joint Research Center for Probiotics and Gut Health, Jiangnan University, Wuxi 214122, China; 4(Yangzhou) Institute of Food Biotechnology, Jiangnan University, Yangzhou 225004, China; 5National Engineering Center of Functional Food, Jiangnan University, Wuxi 214122, China; 6Wuxi Translational Medicine Research Center and Jiangsu Translational Medicine Research Institute Wuxi Branch, Wuxi 214122, China

**Keywords:** Alzheimer’s disease, gut-brain axis, *Bifidobacterium*, cognitive impairment, microbiota

## Abstract

Psychobiotics are used to treat neurological disorders, including mild cognitive impairment (MCI) and Alzheimer’s disease (AD). However, the mechanisms underlying their neuroprotective effects remain unclear. Herein, we report that the administration of bifidobacteria in an AD mouse model improved behavioral abnormalities and modulated gut dysbiosis. *Bifidobacterium breve* CCFM1025 and WX treatment significantly improved synaptic plasticity and increased the concentrations of brain-derived neurotrophic factor (BDNF), fibronectin type III domain-containing protein 5 (FNDC5), and postsynaptic density protein 95 (PSD-95). Furthermore, the microbiome and metabolomic profiles of mice indicate that specific bacterial taxa and their metabolites correlate with AD-associated behaviors, suggesting that the gut–brain axis contributes to the pathophysiology of AD. Overall, these findings reveal that *B. breve* CCFM1025 and WX have beneficial effects on cognition via the modulation of the gut microbiome, and thus represent a novel probiotic dietary intervention for delaying the progression of AD.

## 1. Introduction

Alzheimer’s disease (AD) is a complex progressive condition, with symptoms that include amyloid plaques, neurofibrillary tangles, cognitive impairment, neurodegeneration, and neuroinflammation [1,2,3]. Despite the existence of treatments that can alleviate these symptoms, no therapeutic approach has been proven to completely halt disease progression [4]. Thus, there is a pressing need to identify novel strategies, such as combining dietary interventions known to improve general brain function with anti-Alzheimer’s therapy [5].

In the past decade, emerging studies have suggested that gut microbiota plays a role in the modulation of brain function and behavior. However, these studies were mainly carried out in animal models [6,7,8]. The gut microbiota–brain axis is a bidirectional communication pathway connecting the gut and brain, and the regulation of this axis, particularly via dietary supplementation, has been proposed as a novel therapeutic approach for neurodevelopmental disorders such as AD, depression and anxiety [9,10,11].

Psychobiotics, a class of probiotics, are live organisms that are thought to benefit patients with psychiatric illnesses [12]. Psychobiotics can produce and deliver neuroactive substances, thereby affecting the gut–brain axis [13]. Previous studies have reported that the supplementation of AD mice with probiotics (*Lactobacillus* and *Bifidobacterium*) enhanced learning capacity, significantly reversed impaired alternation behavior, and partially ameliorated cognitive decline via the release of neurotransmitters and the bacterial metabolite acetate [14,15]. Similarly, several clinical studies have indicated that probiotic administration improves cognitive function and metabolic disorders in patients with AD and mild cognitive impairment (MCI) [16,17,18]. In addition, a clinical study revealed that multispecies probiotic interventions influence serum tryptophan metabolism and alter the composition of the gut microbiome in AD patients [19]. Taken together, these observations indicate that targeting AD-induced gut dysbiosis may be an effective AD therapy.

The present study aimed to investigate the association between bifidobacteria, the gut microbiome, bacterial metabolites, and AD, as well as to further examine potential intervention methods. To this end, we established an animal model of AD and compared the effects of five *Bifidobacterium breve* strains on cognitive function, neuroinflammation, gut microbial community structure, and host metabolites in Aβ_1-42_-treated mice. We show that the administration of either *B. breve* CCFM1025 or WX improved cognitive function in Aβ_1-42_-treated mice via the modulation of the gut microbiome.

## 2. Materials and Methods

### 2.1. Bacterial Treatment

The five strains of *B. breve* (*B. breve* NMG, *B. breve* MY, *B. breve* CCFM1025, *B. breve* XY, and *B. breve* WX) used in this study were isolated from fecal samples of healthy human subjects of various ages (shown in Table 1) and stored at the culture collection of food microorganisms in Jiangnan University (Wuxi, Jiangsu, China). *B. breve* JSWX22M4 (WX) has been preserved at the Food Biotechnology Center of Jiangnan University under the respective serial numbers CCFM1179. All the participants provided written informed consent for the collection of their fecal samples for research purposes. All bacteria strains were cultured in modified de Man, Rogosa and Sharpe (MRS) broth supplemented with 0.05% *w/v* L-cysteine-, and incubated at 37 °C under anaerobic conditions (Electrotek 400TG, West Yorkshire, UK).

The bacteria cells were collected by centrifugation and suspended in 10% skimmed milk to yield a final concentration of 10^9^ colony-forming units (CFU) per milliliter for oral administration.

### 2.2. Beta-Amyloid (_1-42_) Preparation

Amyloid-β (Aβ) protein (Aβ_1-421-42_, cat. no. H-1368) was purchased from Bachem (Bachem, Bubendorf, Switzerland). The dissolution of the Aβ_1-42_ peptide was performed according to the manufacturer’s instructions. Aβ_1-42_ oligomers were obtained using a well-established protocol with some modifications [20].

### 2.3. Animals

Eighty 8-week-old male C57BL/6J mice were purchased from the Model Animal Research Center of Nanjing University (Nanjing, China). All the mice were kept in the Animal Center of Jiangnan University under environmentally controlled conditions (22 °C ± 3 °C, 12-hour light/dark cycle). They had free access to a standard chow diet and sterile water. All experimental procedures were approved by the Animal Experimentation Ethics Committee of Jiangnan University (qualified number: JN.No20190415c0800618(74)).

After acclimatization for seven days, 64 mice were randomly divided into 8 groups (*n* = 8 per group). As a sham operation group, mice in the control group received an intrahippocampal injection of phosphate-buffered saline (PBS). However, mice in the other seven groups (including model, donepezil, and five *B. breve* groups) all received an intrahippocampal injection of 1 μL Aβ1-42 oligomer. Table 2 presents details of the treatment. As for the positive medicine control group, the dose of donepezil was set as 3 mg/kg body weight per day. Donepezil and *B. breve* intake was started one week after the Aβ_1-42_ injection and continued for 6 weeks. The experimental procedure timeline is shown in Figure 1.

### 2.4. Surgical Procedure 

Stereotaxic injections of Aβ_1-42_ were performed as described previously [2]. Mice were anesthetized by the inhalation of 3% isoflurane and maintained with 1% isoflurane, and then placed in a stereotaxic apparatus to receive an intrahippocampal injection of 1 μL Aβ_1-42_ oligomer or phosphate-buffered saline (PBS) at these coordinates: anterior-posterior, −2.0 mm; midline-lateral, ±1.8 mm; dorsal-ventral, −2.0 mm from the bregma. The injection was performed using a 5 μL Gaoge syringe attached to a digital stereotaxic apparatus and the rate of infusion was 0.15 μL/min. After the infusion was completed, the needle remained in place for 10 min before slow withdrawal [21].

### 2.5. Behavior Tests

Behavior tests were started after the *Bifidobacterium* strain intervention, and mice were allowed to rest between tests. We performed three different behavioral tests: a Y-maze to evaluate short-term spatial working memory, a Morris water maze (MWM) to measure spatial reference memory, and a passive avoidance test to measure learning and memory. Behavior analyses were performed blindly during the dark phase from 8:00 to 15:00.

#### 2.5.1. Y-Maze

The Y-maze used in this study did not involve any training, reward, or punishment. During a 5-min session, each mouse was placed in the center of the symmetrical Y-maze and was allowed to explore freely. An alternation was considered as consecutive entry into all three arms. The number of maximum alternations and the percentage of alternation were calculated as described previously [2].

#### 2.5.2. Morris Water Maze

Each trial of the Morris water maze consisted of an acquisition phase (5 consecutive learning days) and a probe phase (1 probe day). During the spatial acquisition phase, mice were given four training trials using four start locations each day. The mouse was released and stopped when it touched the platform. During the probe phase (day 6), the platform was removed. All the mice started from a novel position and trailed for a single period of 60 s. The time taken to find the platform (escape latency) was the parameter we used to evaluate memory. Swimming trajectories were recorded using the Ethovision 11.5 automated tracking system (Noldus, Wageningen, The Netherlands).

#### 2.5.3. Passive Avoidance Test

The apparatus chamber (Ugo Basile, Milan, Italy) used in this test was composed of a black poorly illuminated compartment and a white illuminated compartment. Briefly, during the training test, the mouse was placed in the white compartment. When the mouse innately crossed to the black compartment, it received a mild electric shock to the foot. During the retention test, the passive avoidance response was evaluated 24 h after the foot shock. The better the retention of memory, the higher the latency value.

### 2.6. Tissue Processing

After behavioral tests, mouse fecal samples were collected. Subsequently, mice were deeply anesthetized with isoflurane, and blood samples were collected. Serum was then collected after centrifugation (3500 rpm for 15 min) and stored at −80 °C until analysis. Brains were isolated and bisected longitudinally. Hippocampal and other tissues were dissected and stored frozen at −80 °C for biochemical studies.

### 2.7. ELISA

The tissues were homogenized in pre-cooled PBS supplemented with protease and phosphatase inhibitor. Then, we centrifuged the samples and collected the supernatants to carry out assays. The measurements of brain-derived neurotrophic factor (BDNF) and transforming growth factor-β1 (TGF-β1) levels were performed using ELISA kits from Elabscience (China, Wuhan). The synapse-associated protein 97 (SAP97), Aβ_1-42,_ synaptophysin (SYP), postsynaptic density 95 (PSD95), and fibronectin type III domain-containing protein 5 (FNDC5) levels were determined using Mlbio ELISA kits according to the manufacturer’s protocol (China, Shanghai). The measurement of serum interleukin 6 (IL-6) was performed using ELISA kits from Mlbio (China, Shanghai).

### 2.8. SCFAs Extraction and Analysis

The concentrations of acetate, propionate, and butyrate in fecal samples were determined by gas chromatography–mass spectrometry (GC–MS) (GCMS-QP2010 Ultra system, Shimadzu Corporation, Japan). In brief, 50 mg fecal samples were soaked in saturated NaCl for 30 min. After adding H_2_SO_4_ (20 μL, 10% (*v*/*v*)) for acidification, diethyl ether (800 μL) was added to the tube and mixed thoroughly. Then, the supernatants were analyzed in the organic phase. The GC-MS operating parameters were listed in a previous study [22].

### 2.9. Microbiome Profiling

At the end of the experiment, fecal pellets were collected, snap-frozen and stored at −80 °C. DNA was extracted from the feces using a FastDNA Spin Kit (MP Biomedical, Irvine, CA, USA). Next, the V3-V4 region of the 16S rRNA gene was amplified using the primers 341F and 806R. Then, the PCR products were purified using the TIANgel Mini Purification Kit (TIANGEN, Beijing, China) and quantified using a Qubit PicoGreen dsDNA Assay Kit (Life Technologies, Carlsbad, CA, USA). Library construction, sequencing, and bioinformation analyses were performed as described in [23], using the IlluminaMiSeq platform.

### 2.10. Quantification and Statistical Analysis

Statistical analysis was performed using GraphPad Prism 8.0 (GraphPad Software, La Jolla, CA, USA). Data were plotted as mean ± standard error of mean (SEM). Differences between groups were assessed using one-way analysis of variance (ANOVA) or the Kruskal–Wallis test with post hoc comparison. Statistical significance was verified through Fisher’s least significant difference (LSD), Dunn’s test, or Welch’s *t*-test, where appropriate. A *p*-value < 0.05 was considered to indicate statistical significance. Multivariate statistical analysis was performed using SIMCA 14.1 (Umetrics, Umea, Sweden). The network correlation between variations was visualized by Cytoscape (version 3.8.2, https://cytoscape.org/, accessed on 11 May 2021).

## 3. Results

### 3.1. Bifidobacteria Supplementation Ameliorates Cognitive Decline in Aβ_1-42_-Treated Mice

In a Y-maze (Figure 2A,B), Aβ_1-42_-treated mice demonstrated less spontaneous alternation behavior and fewer total arm entries than control mice. *B. breve* NMG and CCFM1025 administration led to significant improvements in alternation behavior and increases in total arm entries. However, the administration of the other three *B. breve* strains failed to improve working memory.

In a passive avoidance test (Figure 2C), Aβ_1-42_-treated model mice exhibited a shorter latency period than that of control mice. Intriguingly, mice treated with *B. breve* CCFM1025, XY, or WX exhibited a significantly prolonged latency time compared to that of AD mice, although these changes were variable.

During the 5-day Morris water maze (MWM) training, the escape latency of mice gradually decreased in all groups (Figure 2D). During the probe test (Figure 2E,F), Aβ_1-42_-treated mice were significantly slower to acquire escape latency compared to control mice. Additionally, AD mice exhibited a decreased time spent in the targeted quadrant; however, this was not statistically significant. The acquisition of escape latency was significantly improved in the MY and CCFM1025 treatment groups compared to the AD group. Time spent in the targeted quadrant was statistically increased in the MY, CCFM1025, and WX treatment groups. These findings suggest that *B. breve* administration improves Aβ_1-42_-induced memory defects in mice.

### 3.2. Bifidobacteria Supplementation Reduced Hippocampal Aβ_1-42_ Concentrations

Aβ_1-42_ injection induced significant AD-associated pathological neuroinflammation in the brain, including amyloid deposition and synaptic degeneration. The administration of CCFM1025, XY, and WX to Aβ_1-42_-treated mice significantly reduced the hippocampal accumulation of Aβ_1-42_ (Figure 3). However, no obvious changes resulted from the NMG or MY treatment of mice when compared to AD mice.

### 3.3. The Effect of Bifidobacteria Treatment on Cytokine Concentrations

As shown in Figure 4A–D, CCFM1025 treatment significantly improved synaptic plasticity and led to increased concentrations of brain-derived neurotrophic factor (BDNF), fibronectin type III domain containing 5 (FNDC5), and postsynaptic density protein 95 (PSD-95). Intriguingly, all bifidobacteria strains, except for MY, increased the concentrations of BDNF; in MY-treated mice, only the synaptophysin (SYP) concentrations were significantly increased. Moreover, WX treatment increased the concentrations of both SYP and PSD95.

Notably, the transforming growth factor beta 1 (TGF-β1) concentrations were significantly increased in WX-treated mice, while no significant differences were seen between the control and AD mice (Figure 4E). Moreover, the Aβ_1-42_-treated mice exhibited decreased serum concentrations of cytokine interleukin 6 (IL-6) compared to the control mice. NMG and CCFM1025 treatment significantly increased the serum concentrations of IL-6, and these concentrations were comparable to those of the control group (Figure 4F). To identify metabolically beneficial strains, we performed multivariate statistical analyses. Two-dimensional loading plots of principal component analyses indicated that CCFM1025, XY, and WX were located closer to the control group, indicating the ability of these strains to alleviate AD (Figure 5A,B).

### 3.4. Bifidobacteria Administration Altered the Gut Microbiome Composition in Aβ_1-42_-Treated Mice

A higher α-diversity was seen in the bifidobacteria intervention group; however, no statistically significant difference was observed between the AD mice and the control group mice (Figure 6A,B). Significant compositional differences in β-diversity were identified using principal coordinate analysis plots based on Bray–Curtis dissimilarities; however, this was not recapitulated using the weighted UniFrac method (Figure 6C). Remarkably, there was greater variability in the stool microbial composition between groups at the phylum and family levels (Figure 6D,E).

Fifteen genera were classified by random forest and the linear discriminant analysis effect size (LEfSe) (Figure 7A,B). At the genus level, *Desulfovibrio* markedly decreased in abundance in the donepezil-treated group (Figure 7C). XY and WX administration significantly increased the relative abundance of *Akkermansia* and decreased the relative abundance of *Coprococcus* (Figure 7D,F). Moreover, XY and CCFM1025 treatment dramatically increased the *Bifidobacterium* abundance (Figure 7E).

The classification of species using LEfSe identified *B. adolescentis* as the top hit, followed by *Lactobacillus reuteri* and *Akkermansia muciniphila*. The treatment with *B. breve* XY increased the abundance of *B. adolescentis*, which was barely detectable in the Aβ_1-42_-treated mice (Figure 7H). The CCFM1025 and XY treatment normalized the ratio of *L. reuteri*, whereas WX treatment did not significantly alter this ratio compared to the model mice (Figure 7I). Notably, XY and WX treatment significantly enriched the abundance of *A. muciniphila* (Figure 7J).

### 3.5. Alterations in Microbial Metabolites Reveal Potential Mechanisms Underlying AD

We first focused on short-chain fatty acids (SCFAs), which have previously been reported to be associated with AD. Gas chromatography–mass spectrometry (GC–MS) analysis of stool samples of AD mice revealed a significant decrease in butyrate and acetate concentrations, while propionate concentrations were increased (Figure 8A). Treatment with any of the three *B. breve* strains or donepezil treatment significantly increased the concentration of acetate. Butyrate concentrations were markedly increased in the stools of the CCFM1025-treated group. However, only donepezil treatment significantly increased the concentration of propionate.

To further elucidate the contribution of specific microbes and their metabolites to cognitive function in AD mice, we used a correlation network to identify additional microbiome/metabolite neurofunction associations. Correlation analyses suggested that many behavioral and neurological symptoms were differentially affected by specific microbiota and their production of SCFAs (Figure 8B). The abundance of acetate-producing bacteria, such as *Turicibacter* and *Oscillospira*, displayed a positive correlation with the levels of FNDC5 and BDNF. Notably, the level of Aβ_1-42_ was negatively correlated with butyrate concentrations. While this correlational approach allowed us to identify potential drivers of fecal microbiota-induced cognitive improvements, it could not identify specific microbes that may be most relevant for proper brain function.

## 4. Discussion

Although many studies suggest the use of probiotics as a gut microbiota-targeted strategy to combat AD, the exact mechanism underlying the probiotic-mediated amelioration of AD is not clearly understood. In this study, we revealed that *B. breve* exhibits strain-specific effects to alleviate AD comorbidities. *B. breve* CCFM1025 and WX had particularly marked effects on alleviating cognitive impairment and neuroinflammation via the gut microbiota–brain axis during AD progression.

The intrahippocampal injection of Aβ_1-42_ into mice generates one of the most useful animal models of AD, as it can lead to cognitive deficits that resemble AD [24]. We therefore established an Aβ_1-42_-treated mouse model and evaluated the protective effects of five *B. breve* strains against cognitive impairment using a series of behavioral assays. Compared to the control mice, Aβ_1-42_-treated mice developed AD hallmarks, including cognitive deficits, reduced alternation behavior and total arm entries in a Y-maze, a reduced latency time in passive avoidance tests, and a slower acquisition of escape latency in MWM tasks (Figure 2). By contrast, treatment with CCFM1025 or MY improved the outcomes for these mice in all three behavioral tests, consistent with the results of previous studies using *Lactobacillus* and *Bifidobacterium* probiotic strains that were tested both individually and as cocktails [14,17,18,25].

Given that Aβ deposition triggers neuronal dysfunction and cell death in the brain [4], we found that MY and CCFM1025 treatment significantly reduced the deposition of hippocampal Aβ_1-42_. Interestingly, mice in the XY group who exerted no obvious improvements in the behavioral tests also experienced reduced Aβ deposition. This finding suggests that reducing Aβ deposition concentrations alone does not necessarily lead to improved cognition in Aβ_1-42_-treated mice.

With the increasing knowledge of the underlying mechanisms of AD, research has shifted from focusing on Aβ-driven pathologies to studying the role of neuroinflammatory factors during disease progression [26,27]. Probiotics can increase the concentrations of BDNF, growth factors and synaptic markers, and reduce neuroinflammation [28]. Thus, we next measured the concentrations of 12 different substances in the hippocampus of Aβ_1-42_-treated mice with or without *B. breve* intervention. We found that all *B. breve* strains, except MY, variably increased the concentrations of BDNF. BDNF is essential for synaptic plasticity, cognitive function, and learning [29]. Elevating concentrations of BDNF is therefore a mechanism by which bifidobacteria may prevent cognitive impairment in Aβ_1-42_-treated mice. FNDC5 can stimulate hippocampal BDNF expression in mice [30]. The synaptic proteins PSD95 and SAP97 are also important regulators of synaptic plasticity. Given that CCFM1025 treatment increased the concentrations of BDNF, PSD95, and FNDC5, this may explain the improvements seen in cognitive function.

IL-6 has been reported to promote improved cognition [31]. In the current study, IL-6 concentrations were elevated following CCFM1025 or NMG administration. TGF-β1 concentrations are known to increase following brain damage, and can protect against cell death and neurodegeneration [32]. Among the various cytokines present in the hippocampal tissue homogenates, the concentrations of TGF-β1 significantly increased after supplementation with three *B. breve* strains compared to control mice, suggesting that *B. breve* may suppress neuroinflammation.

Although our understanding of neurological disorders such as Parkinson’s disease is increasing, the link between neuroinflammation and the gut microbiome in AD remains unclear [33]. An understanding of how the gut–brain axis influences cognition may provide an effective strategy to delay the progression of AD via microbiota-mediated therapies. Given that AD patients show altered gut microbiomes [34,35,36,37], we sought to determine whether *B. breve* strains affect cognitive function.

β-diversity analyses revealed significant changes to the composition of the gut microbiome following *B. breve* treatment. Emerging evidence indicates that microbiota Bifidobacterium spp. abundances are decreased in various central nervous system disorders, including depression, AD, Parkinson’s disease, and autism spectrum disorder [38,39]. In line with this, a reduction in Bifidobacterium spp. was seen in Aβ_1-42_-treated mice, while mice treated with any of the *B. breve* strains exhibited significantly increased abundances of Bifidobacterium spp. *Desulfovibrio* is a sulfate-reducing bacterium and can produce neurotoxic compounds (e.g., hydrogen sulfide). Previous studies have reported that *Desulfovibrio* is enriched in the microbiota of those with autism or depression, but not in the microbiota of those with AD [40]. Notably, our data indicate that the relative abundance of *Desulfovibrio* (Deferribacteres) markedly decreased after donepezil treatment. Intriguingly, probiotic administration was previously shown to normalize the abundance of *Desulfovibrio* in the microbiota of those with depression [40]; however, we observed no significant difference in the abundance of *Desulfovibrio* in the microbiota of mice treated with *B. breve*.

Given that different species from the same genus may respond differently to the same intervention, it is necessary to identify responses to *B. breve* treatment at the species level. By further using LEfSe to classify the most specific species, we found that *B. adolescentis* was the top hit, followed by *L. reuteri* and *A. muciniphila*. Due to their beneficial effects on human health, *B. adolescentis* and *L. reuteri* are widely used as probiotics. Animal studies have previously reported that *B. longum* CCFM687 supplementation can prevent the onset of depression due to chronic stress [28], while *B. breve* CCFMCCFM1025 administration was found to exert antidepressant-like effects [39]. Randomized controlled trials have also demonstrated that *Bifidobacterium* intervention improved cognitive decline in those with MCI and severe dementia [18,25]. Taken together with the improved cognitive function we observed for specific *B. breve* strains, large-scale, multi-center clinical trials may be warranted in the future to identify the specific effect of probiotics on AD patients at different stages of the disease, particularly in the context of the early onset of AD.

In addition to the effects of the gut microbiome on cognition, bacterial metabolites can also modulate the activity of the host brain. Previous studies have suggested that there is a dysregulation of gut metabolites—including lipopolysaccharide, SCFAs, bile acids and amino acids—in psychiatric diseases [41,42,43]. Here, we quantified the abundance of the major SCFAs with gut microbial fermentative activity: acetate, propionate and butyrate. The stool samples of Aβ_1-42_-treated mice showed a significant reduction in SCFA concentrations. The concentration of acetate was dramatically increased in mice treated with any of the *B. breve* strains or donepezil. Butyrate concentrations were also significantly increased in the CCFM1025 group. Moreover, correlation analysis suggested that many behavioral and neurological symptoms were differentially affected by SCFAs. However, the mechanisms by which SCFAs regulate Aβ-induced pathophysiology are likely to be complex. Additional studies are warranted to explore these aspects, as it appears that SCFAs stimulate the production of neurotransmitters, which exert strong anti-inflammatory effects via the gut–brain axis [44].

This study has several limitations. As the Aβ_1-42_-treated mouse model might not fully represent the main pathological hallmarks of human AD, future clinical studies are necessary to explore the effect of *B. breve* strains on cognitive function in AD. In addition, although we have revealed a potential role of gut dysbiosis in AD progression, it will be critical to identify the key taxa responsible for neuroinflammation. Moreover, based on the complementary or synergistic effect of probiotic supplementation on the gut microbiota, future studies are required to compare the effects of different probiotic combinations on AD and formulate optimal multi-strain probiotic interventions.

## 5. Conclusions

Overall, we have revealed the ability of *B. breve* treatment to alleviate cognitive dysfunction and delay AD progression in mice. The mechanisms underlying the neuroprotective effects of this treatment may be related to neuroinflammation and cognition pathways, possibly via the promotion of neurotransmitter production (BDNF) and the regulation of the gut microbial composition. Taken together, our findings suggest that it may be possible to design a novel microbiota-based probiotic dietary intervention to prevent or alleviate the symptoms of AD and other forms of dementia.

## Figures and Tables

**Figure 1 nutrients-13-01602-f001:**
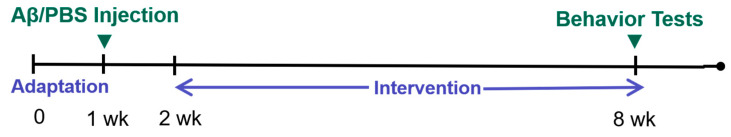
The experimental procedure timeline. Note: wk = week, PBS = phosphate-buffered saline.

**Figure 2 nutrients-13-01602-f002:**
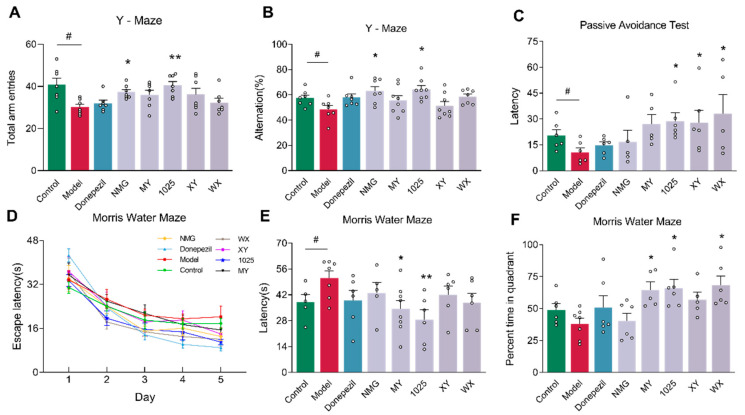
*Bifidobacterium breve* supplementation impacts the cognitive function and behaviors of mice. (**A**) Total arm entries in Y-maze test (*n* = 6–8). (**B**) Spontaneous alternation behavior in Y-maze test (*n* = 6–8). (**C**) Escape latency of passive avoidance test (*n* = 6–8). (**D**) Mean escape latency to reach the platform in Morris water maze training trials (*n* = 6–8). Mice were given four trials per day, and data represent the mean ± standard error of the mean (mean ± SEM) of four trials. (**E**) The escape latency during the probe phase (day 6) of the Morris water maze (*n* = 6–8). (**F**) Percentage of time in the target quadrant during the probe phase of the Morris water maze (*n* = 6–8). Data are presented as mean ± standard error of the mean (mean ± SEM). Control vs. model: # *p* < 0.05 by unpaired student’s *t*-test; * *p* < 0.05, ** *p* < 0.01 by one-way ANOVA for all groups.

**Figure 3 nutrients-13-01602-f003:**
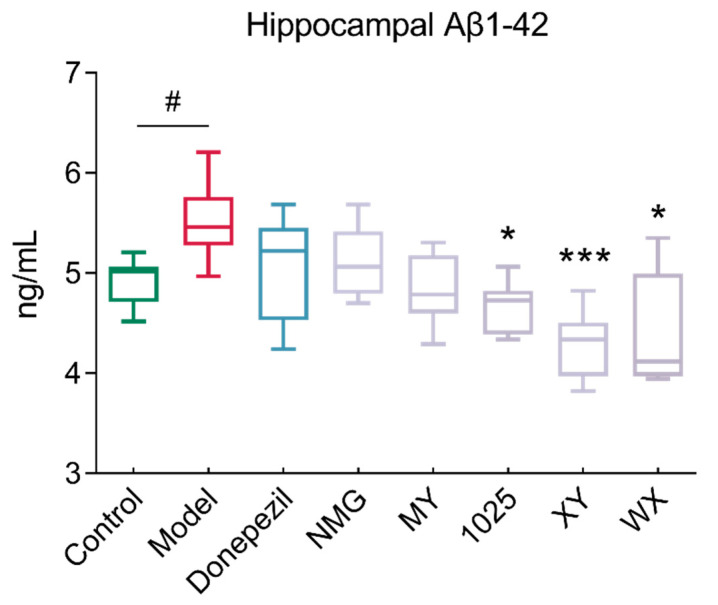
Changes in Aβ1-42 levels in the hippocampal homogenates of mice in different groups (*n* = 6–8). In the box plot, the bottom and top are, respectively, the 25th and 75th percentile; a line within the box marks the median. Whiskers above and below the box indicate the 1.5 interquartile range of the lower and upper quartile, respectively. Control vs. Model: # *p* < 0.05 by unpaired student’s *t*-test; * *p* < 0.05, *** *p* < 0.001 by one-way ANOVA for all groups.

**Figure 4 nutrients-13-01602-f004:**
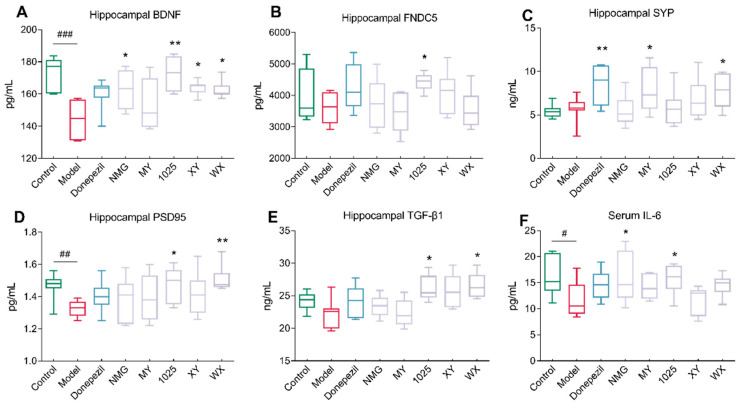
Effect of *Bifidobacterium breve* treatment on brain, neuroinflammation, and synaptic function. (**A**) Hippocampal BDNF levels (*n* = 6–8). (**B**) Hippocampal FNDC5 levels (*n* = 6–8). (**C**) Hippocampal SYP levels (*n* = 6–8). (**D**) Hippocampal PSD95 levels (*n* = 6–8). (**E**) Hippocampal TGF-β1 levels (*n* = 6–8). (**F**) Serum IL-6 levels (*n* = 6–8). In the box plot, the bottom and top are, respectively, the 25th and 75th percentiles, and a line within the box marks the median. Whiskers above and below the box indicate the 1.5 interquartile range of the lower and upper quartiles, respectively. Control vs. Model: # *p* < 0.05, ## *p* < 0.01, ### *p* < 0.001 by unpaired student’s *t*-test; * *p* < 0.05, ** *p* < 0.01 by one-way ANOVA for all groups.

**Figure 5 nutrients-13-01602-f005:**
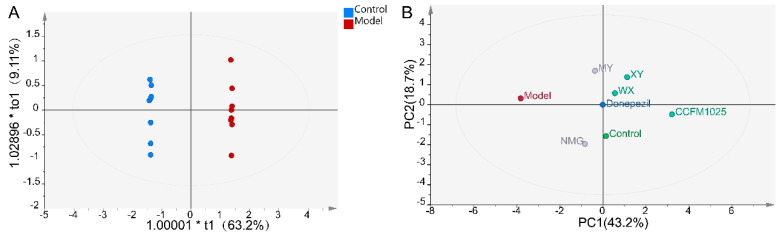
Statistical analysis of all significant measurements. (**A**) OPLS-DA plot for the control and model groups (*n* = 6–8). (**B**) PCA score plot for all groups. The distance between two data points in the two-dimensional PCA plot is a visual representation of the similarity between different groups. Note: PCA = principal component analyses, PC1 = principal component 1, PC2 = principal component 2.

**Figure 6 nutrients-13-01602-f006:**
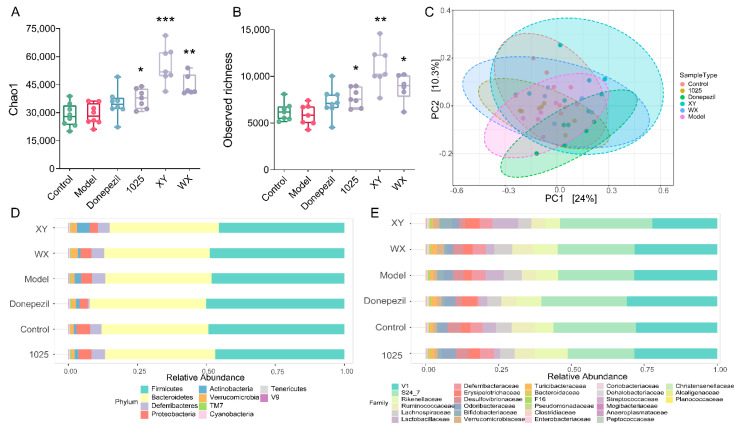
*Bifidobacterium breve* supplementation re-shaped the gut microbiome. (**A**,**B**) α-diversity was measured by Chao1 (left) and observed richness (right) (*n* = 6–8). In the box plot, the bottom and top are, respectively, the 25th and 75th percentiles, and a line within the box marks the median. Whiskers above and below the box indicate the 1.5 interquartile range of the lower and upper quartiles, respectively. * *p* < 0.05, ** *p* < 0.01, *** *p* < 0.001 by one-way ANOVA for all groups. (**C**) Principal coordinate analysis (PCoA), based on Bray–Curtis distance, and PERMANOVA was used to test the difference in gut microbiota composition and diversity (*p* < 0.001, R^2^ = 0.29; *n* = 6–8 mice/group). (**D**) The microbial distribution of bacteria at the phylum level (*n* = 6–8). (**E**) The microbial distribution of bacteria at the family level (*n* = 6–8).

**Figure 7 nutrients-13-01602-f007:**
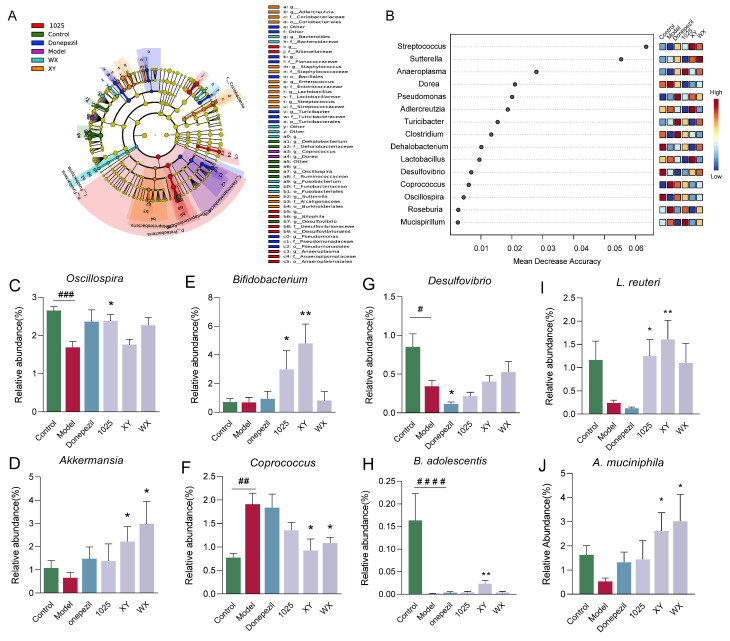
Effect of *Bifidobacterium breve* supplementation on the composition of the gut microbiota. (**A**) Linear discriminant analysis (LDA) effect size (LEfSe). Differential taxa are labeled with tags and annotated in the right panel. Data were computed with an LDA score above 2.00 and *p*-value below 0.05 for the factorial Kruskal–Wallis test. (**B**) Significant features were identified by Random Forest. The features are ranked by the mean decrease in classification accuracy when they are permuted. (**C**–**G**) The relative abundance of selected taxa at the genus level (*n* = 6–8). (**H**–**J**) The relative abundance of selected taxa at the species level (*n* = 6–8). Data are presented as mean ± standard error of the mean (mean ± SEM). Control vs. Model: # *p* < 0.05, ## *p* < 0.01, ### *p* < 0.001, #### *p* < 0.0001 by unpaired student’s *t*-test; * *p* < 0.05, ** *p* < 0.01 by one-way ANOVA.

**Figure 8 nutrients-13-01602-f008:**
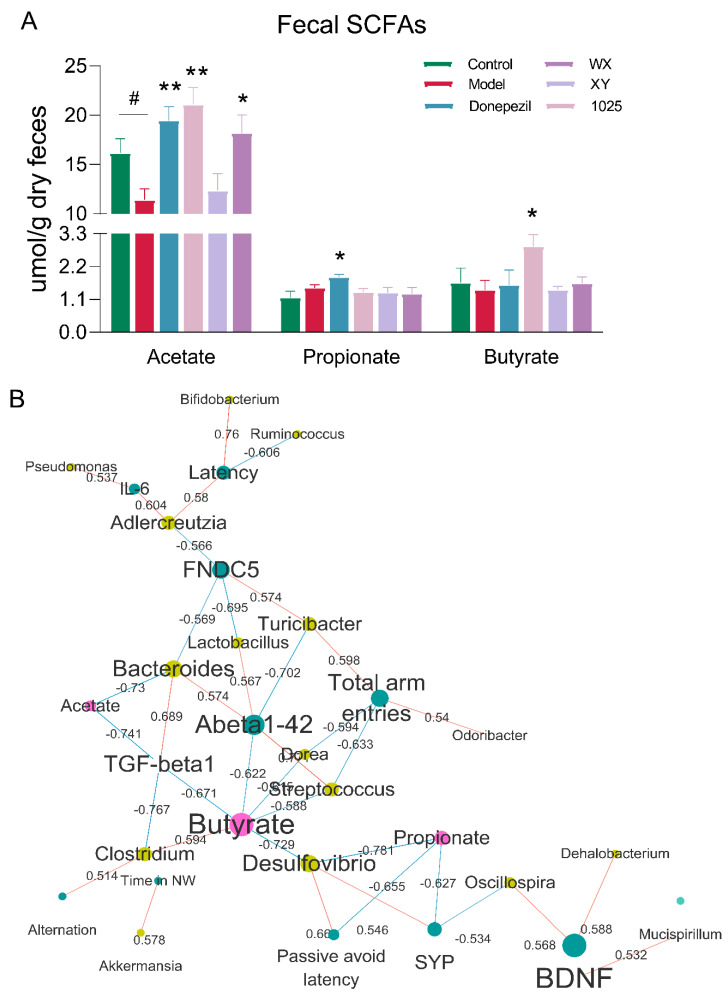
Alterations in microbial metabolites reveal potential mechanisms underlying AD. (**A**) The concentrations of SCFAs (*n* = 6–8). Data are presented as mean ± standard error of the mean (mean ± SEM). Control vs. Model: # *p* < 0.05 by unpaired student’s *t*-test; * *p* < 0.05, ** *p* < 0.01 by one-way ANOVA. (**B**) Network representing significant and stability-selected correlations of cognition (blue nodes) with fecal microbial taxa at the genus level (yellow nodes) and fecal metabolites (red nodes). Red edges indicate a positive correlation and blue edges indicate a negative correlation, while the number on the edge indicates correlation. Note: SCFAs = short chain fatty acids, FNDC5 = fibronectin type III domain-containing protein 5, TGF = transforming growth factor, BDNF = brain-derived neurotrophic factor.

**Table 1 nutrients-13-01602-t001:** Bifidobacteria strains used in this study.

Bifidobacteria Strain	Sample	Gender	Age	Regional Origin
*B. breve* NMGCF1M12 (NMG)	Human feces	Female	8 months	Chifeng, Inner Mongolia, China
*B. breve* HuNMY8M2 (MY)	Human feces	Male	3 months	Mayang, Hunan Province, China
*B. breve* CCFM1025	Human feces	Male	38 years	Zoige County, Sichuan Province, China
*B. breve* HNXY26M4 (XY)	Human feces	Female	12 months	Shangqiu, Henan Province, China
*B. breve* JSWX22M4 (WX)	Human feces	Male	8 months	Wuxi, Jiangsu Province, China

**Table 2 nutrients-13-01602-t002:** The treatment of mice in different groups.

Group	Surgical Injection (Intrahippocampal)	Dietary Intervention
Control	Phosphate-buffered saline (PBS)	200 μL skimmed milk (10%)
Model	Aβ_1-42_ (2 μg/μL)	200 μL skimmed milk (10%)
Donepezil	Aβ_1-42_ (2 μg/μL)	200 μL Donepezil (3 mg/kg/d)
NMG	Aβ_1-42_ (2 μg/μL)	200 μL bacteria suspension (3 × 10^9^ CFU/mL cells)
MY	Aβ_1-42_ (2 μg/μL)	200 μL bacteria suspension (3 × 10^9^ CFU/mL cells)
CCFM1025	Aβ_1-42_ (2 μg/μL)	200 μL bacteria suspension (3 × 10^9^ CFU/mL cells)
XY	Aβ_1-42_ (2 μg/μL)	200 μL bacteria suspension (3 × 10^9^ CFU/mL cells)
WX	Aβ_1-42_ (2 μg/μL)	200 μL bacteria suspension (3 × 10^9^ CFU/mL cells)

## Data Availability

The data that support the findings of this study are available on request from the corresponding author.

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
