# Peer review of "Administration of Bifidobacterium breve Improves the Brain Function of Aβ1-42-Treated Mice via the Modulation of the Gut Microbiome"

_nutrients, 2021, doi:10.3390/nu13051602_

Round 1
Reviewer 1 Report
The study by Zhu et al tested whether the psychobiotic, Bifidobacterium breve, is beneficial for the treatment of Alzheimer’s disease. Using a mouse model of Alzheimer’s disease, the authors found that bifidobacterial improved cognitive function and modulated gut dysbiosis. Overall, this is a comprehensive study with interesting findings. The following comments need to be addressed:
- Methods: Fecal samples were obtained form healthy human subjects. Were the subjects the same sex, age etc?
- Methods: It is stated that 80 mice were divided into 8 groups (n=8 per group). Should this be 64 mice instead?
- Table 1: Did the model and donepezil groups receive Ab1-42 as well? This is not indicated in the table.
- What is the purpose of the donepezil group?
- All figure legends are missing n numbers and statistical tests. Are all data mean ±E.M.?
- Figure 1E: Which day of the trials does this data correspond to?
- Why was serum IL-6 measured and not IL-6 levels in the brain?
- Can the authors measure pro-inflammatory markers in the brain? It would be interesting to see whether Bifidobacterium breve can reduce neuroinflammation.
- Can the authors provide more information on the different strains of Bifidobacterium breve used in the study?
- There are several typos where linking words are missing in sentences.
Author Response
Response to reviewers’ comments
(Manuscript ID: nutrients-1179570, Title: Administration of Bifidobacterium breve improves the brain function of Aβ1-42-treated mice via modulation of the gut microbiome)
Dear Editors and Reviewers,
Thank you for your professional suggestions, which are very helpful for revising and improving our paper, as well as the important guiding significance to our further researches. We have studied comments carefully and revised this manuscript accordingly. In order to make our article more readable, we have added a Table and a Figure in Materials and Methods section in the revised manuscript, so the number of the previous Tables and Figures has changed. Changes in manuscript are highlighted in yellow, and responses to your specific comments are detailed below.
Sincerely yours,
Gang Wang
School of Food Science and Technology
Jiangnan University
Wuxi, Jiangsu, 214122, P.R. China
Tel: (86)510-85912155
Fax: (86)510-85912155
E-mail: wanggang@jiangnan.edu.cn
Comments and Suggestions for Authors:
Reviewer 1:
Comments to the Author
The study by Zhu et al tested whether the psychobiotic, Bifidobacterium breve, is beneficial for the treatment of Alzheimer’s disease. Using a mouse model of Alzheimer’s disease, the authors found that bifidobacterial improved cognitive function and modulated gut dysbiosis. Overall, this is a comprehensive study with interesting findings. The following comments need to be addressed:
Point 1:Methods: Fecal samples were obtained from healthy human subjects. Were the subjects the same sex, age etc?
Response 1: Thank you for your professional suggestion. Actually, the human subjects in this study were of different genders and various ages. According to your advice, we listed the detailed information of the source of the strains in Table 1 and shown below as well. Please see Line 66 and Table 1 in Section 2.1.
Table 1. Bifidobacteria used in this study.
Bifidobacteria strain |
Sample |
Gender |
Age |
Regional origin |
B. breve NMG |
Human faeces |
Female |
8 months |
Chifeng, Inner Mongolia, China |
B. breve MY |
Human faeces |
Male |
3 months |
Mayang, Hunan Province, China |
B. breve CCFM1025 |
Human faeces |
Male |
38 years |
Zoige County, Sichuan Province, China |
B. breve XY |
Human faeces |
Female |
12 months |
Shangqiu, Henan Province, China |
B. breve WX |
Human faeces |
Male |
8 months |
Wuxi, Jiangsu Province, China |
- Methods: It is stated that 80 mice were divided into 8 groups (n=8 per group). Should this be 64 mice instead?
Response 2: Thank you so much for your careful check. Actually, we did use 64 mice in this study and the number mistake has been corrected in Line 89. We feel so sorry for our carelessness.
- Table 1: Did the model and donepezil groups receive Ab1-42 as well? This is not indicated in the table.
Response 3: Yes, mice in the model and donepezil groups received an intrahippocampal injection of 1μL Aβ1-42 oligomer as well. We feel extremely sorry for the typo we should have avoided. Actually, mice in Control group (sham operation group) received an intrahippocampal injection of phosphate-buffered saline (PBS), and mice in other seven groups (including Model, Donepezil and five B. breve groups) all received an intrahippocampal injection of 1μL Aβ1-42 oligomer. We have corrected it in Table 2 and added the detailed information in Section 2.3. Please see the revised parts in Line 90-95. Thanks again for your careful check.
- What is the purpose of the donepezil group?
Response 4: Thank you for your professional suggestion. Donepezil is one of the four medicines used for the treatment of Alzheimer's disease[1,2]. In our study, the donepezil group was used as the positive medicine control group. We have added the purpose of the donepezil group in Line 93-95.
- All figure legends are missing n numbers and statistical tests. Are all data mean ±E.M.?
Response 5: According to your kind and professional suggestion, we added the n numbers, statistical tests and the form of data presented in all Figure legends. Please see the revised legends in the figure legends of Figure 2-8.
- Figure 1E (Figure 2E in revised manuscript): Which day of the trials does this data correspond to?
Response 6: Thank you for your professional comment. Since each trial of the morris water maze consisted of an acquisition phase (5 consecutive learning days) and a probe phase (1 probe day), Figure 2E presented the escape latency during probe phase of morris water maze. In other words, Figure 2E visualized the data in the 6th day of the morris water maze test. We have added the detailed information in the legend of Figure 2E. Please see Line 201-202.
- Why was serum IL-6 measured and not IL-6 levels in the brain?
Response 7: We totally understand the reviewer’s concern. Interleukin-6 (IL-6) is regarded as a proinflammatory cytokine and the serum IL-6 concentration plays a central role in inflammation with age[3]. Moreover, IL-6 has also been reported to benefit cognition and regulate neurogenesis[4]. Given the limited hippocampal tissue and the fact that IL-6 can cross the blood brain barrier[5], we only measured the serum IL-6 in this work. We totally agree with your professional suggestion and will measure the concentration of IL-6 both in serum and the brain in our future study according to your advice. Thanks again for your valuable comment.
- Can the authors measure pro-inflammatory markers in the brain? It would be interesting to see whether Bifidobacterium breve can reduce neuroinflammation.
Response 8: We totally agree with you that it would be important and challenging to explore whether Bifidobacterium breve can reduce neuroinflammation. Researchers have demonstrated that probiotics can increase the concentrations of growth factors and synaptic markers, and can reduce neuroinflammation[6,7]. Unfortunately, in this study, due to the limited hippocampal tissue, we did not detect the inflammation indicators in hippocampus. In fact, we plan to further investigate the roles of the microglia and immune cells during AD development in our future study. For example, as a pro-inflammatory marker, the brain-infiltrated peripheral T helper 1 (Th1) cells are associated with the M1 microglia activation, contributing to AD-associated neuroinflammation[8]. Given the well-recognized functional crosstalk between gut-brain axis[9], investigating the differentiation and proliferation of pro-inflammatory Th1 cells will help us better understand the underlying mechanisms of AD. Thanks again for your profession comment, which will guide our further researches.
- Can the authors provide more information on the different strains of Bifidobacterium breve used in the study?
Response 9: Thank you for your valuable suggestion. We have added the detailed information of the strains used in this study in Table 1. Please see Line 66 and Table 1 in Section 2.1. In addition, B. breve CCFM1025 showed considerable antidepressant-like and microbiota regulating effects in previous studies[10].
- There are several typos where linking words are missing in sentences.
Response 10: Thank you for your valuable comments. We have made every effort to improve the quality and clarity of the language throughout the manuscript. Changes that have been made to the manuscript were highlighted in yellow. Please see Line 58, 125, 144-145, 158, 163, 164, 185, 195, 208-210, 235, 252, 264-275, 301, 367-386.
Reference:
- Scheltens, P.; Blennow, K.; Breteler, M.M.B.; de Strooper, B.; Frisoni, G.B.; Salloway, S.; Van der Flier, W.M. Alzheimer's disease. Lancet 2016, 388, 505-517, doi:10.1016/s0140-6736(15)01124-1.
- Waldemar G, Xu Y, Mackell J. The eff ects of donepezil on dichotomous milestones in patients with Alzheimer’s disease. Int Psychogeriatr 2009; 21: 205.
- Rasmussen P , Vedel J C , Olesen J , et al. In humans IL-6 is released from the brain during and after exercise and paralleled by enhanced IL-6 mRNA expression in the hippocampus of mice[J]. Acta Physiologica, 2015, 201(4):475-482.
- Baier, P.C.; May, U.; Scheller, J.; Rose-John, S.; Schiffelholz, T. Impaired hippocampus-dependent and -independent learning in IL-6 deficient mice. Behavioural Brain Research 2009, 200, 192-196.
- Banks, W.A., Kastin, A.J. & Broadwell, R.D. 1995. Passage of cytokines across the blood-brain barrier. Neuroimmunomodulation 2, 241–248
- Heneka, M.T.; Carson, M.J.; El Khoury, J.; Landreth, G.E.; Brosseron, F.; Feinstein, D.L.; Jacobs, A.H.; Wyss-Coray, T.; Vitorica, J.; Ransohoff, R.M., et al. Neuroinflammation in Alzheimer's disease. Lancet Neurology 2015, 14, 388-405, doi:10.1016/s1474-4422(15)70016-5.
- Tian, P.; Zou, R.; Song, L.; Zhang, X.; Jiang, B.; Wang, G.; Lee, Y.K.; Zhao, J.; Zhang, H.; Chen, W. Ingestion of Bifidobacterium longum subspecies infantis strain CCFM687 regulated emotional behavior and the central BDNF pathway in chronic stress-induced depressive mice through reshaping the gut microbiota. Food & Function 2019, 10.
- Wang, X.; Sun, G.; Feng, T.; Zhang, J.; Geng, M. Sodium oligomannate therapeutically remodels gut microbiota and suppresses gut bacterial amino acids-shaped neuroinflammation to inhibit Alzheimer's disease progression. Cell Research 2019, 29, 1-17.
- Schetters, S. T. T., Gomez-Nicola, D., Garcia-Vallejo, J. J. & Van Kooyk, Y. Neuroinflammation: microglia and T cells get ready to tango. Front. Immunol. 8, 1905 (2017).
- Tian, P.; O'Riordan, K.J.; Lee, Y.K.; Wang, G.; Chen, W. Towards a psychobiotic therapy for depression: Bifidobacterium breve CCFM1025 reverses chronic stress-induced depressive symptoms and gut microbial abnormalities in mice. Neurobiology of Stress 12.

Reviewer 2 Report
The study proposed by Dr Zhu and colleagues investigates the effects of bifidobacteria administration in mice that received intrahippocampal injections of amyloid beta. The conclusions reached by the authors are that two strains of bifidobacterium tested (i.e. B. breve 1025 and WX) show “marked effects to alleviate cognitive impairment and neuroinflammation via the gut-microbiota-brain-axis during AD progression”.
Indeed, these conclusions are not clearly supported by the data presented and, overall, there are important limits and inaccuracies.
- Page 2. In lines 88-89 is reported: 80 mice were divided into 8 groups (n = 8 per group). In fact, that is 64 and not 80.
- Page 3. From Table 1 it could be argued that the Model group did not receive the amyloid injection, thus defeating the entire study. The same mistake occurs in the Donepezil group.
- Page 3, in line 103 is reported “Behavior tests were started at the eighth week”, but it is not specified whether it means at the eighth week after amyloid injection or after bifidobacterium administration.
- Page 4, in line 137 is written that tissues were homogenized in PBS containing protease and phosphatase inhibitors. However, protein extraction from tissues in the absence of detergents (such as SDS, Triton etc.) is a somewhat unusual procedure.
- Figure 2. In the legend there is only the title. The symbols of statistical significance are not consistent with the results described in the text (see page 5, line 201-202).
- Figure 3. While BDNF, FNDC5, SYP, PSD-95, and TGF-beta1 have been measured in hippocampal tissue, IL-6 was measured in serum. Why?
- Although the extent of the differences found between treatments is not specified for any experiment, nor is the meaning of the symbols used for statistics, in many cases the effects (albeit statistically significant) are very small. This limits the impact of the results and the validity of the conclusions drawn.
Author Response
Response to reviewers’ comments
(Manuscript ID: nutrients-1179570, Title: Administration of Bifidobacterium breve improves the brain function of Aβ1-42-treated mice via modulation of the gut microbiome)
Dear Editors and Reviewers,
Thank you for your professional suggestions, which are very helpful for revising and improving our paper, as well as the important guiding significance to our further researches. We have studied comments carefully and revised this manuscript accordingly. In order to make our article more readable, we have added a Table and a Figure in Materials and Methods section in the revised manuscript, so the number of the previous Tables and Figures has changed. Changes in manuscript are highlighted in yellow, and responses to your specific comments are detailed below.
Sincerely yours,
Gang Wang
School of Food Science and Technology
Jiangnan University
Wuxi, Jiangsu, 214122, P.R. China
Tel: (86)510-85912155
Fax: (86)510-85912155
E-mail: wanggang@jiangnan.edu.cn
Comments and Suggestions for Authors:
Reviewer 2
Comments to the Author:
The study proposed by Dr Zhu and colleagues investigates the effects of bifidobacteria administration in mice that received intrahippocampal injections of amyloid beta. The conclusions reached by the authors are that two strains of bifidobacterium tested (i.e. B. breve 1025 and WX) show “marked effects to alleviate cognitive impairment and neuroinflammation via the gut-microbiota-brain-axis during AD progression”.
Indeed, these conclusions are not clearly supported by the data presented and, overall, there are important limits and inaccuracies.
Response: We gratefully thanks for the precious time you spent making constructive and professional comments. These comments are all valuable and very helpful for revising and improving our manuscript, as well as the important guiding significance to our further researches. We have studied comments carefully and have made corrections which we hope meet with approval.
- Page 2. In lines 88-89 is reported: 80 mice were divided into 8 groups (n = 8 per group). In fact, that is 64 and not 80.
Response 1: Thank you so much for your careful check. Actually, we did use 64 mice in this study and the number mistake has been corrected in Line 89. We feel so sorry for our carelessness.
- Page 3. From Table 1 it could be argued that the Model group did not receive the amyloid injection, thus defeating the entire study. The same mistake occurs in the Donepezil group.
Response 2: We feel extremely sorry for the typos we should have avoided. To compare the effects of five B. breve strains administration on cognitive function, we established an AD mouse model. Actually, mice in Control group (sham operation group) received an intrahippocampal injection of phosphate-buffered saline (PBS), and mice in other seven groups (including Model, Donepezil and five B. breve groups) all received an intrahippocampal injection of 1μL Aβ1-42 oligomer. We have corrected it in Table 2 and added the detailed information in Section 2.3. Please see the revised parts in Line 90-95. Thank you again for your careful check.
- Page 3, in line 103 is reported “Behavior tests were started at the eighth week”, but it is not specified whether it means at the eighth week after amyloid injection or after bifidobacterium administration.
Response 3: We feel sorry for the inconvenience brought to the reviewer. By saying “Behavior tests were started at the eighth week”, we mean “at the eighth week of the animal experiment”. We have specified this sentence and added the experimental procedures timeline as the Figure 1 in Section 2.3 as well as below. Please see Line 96 & 110.
Figure 1: The experimental procedures timeline.
After adaptation for seven days, mice received an intrahippocampal injection of PBS or Aβ1-42. Dietary intervention was started one week after the Aβ1-42 injection and continued for 6 weeks. Behavior tests were started at the end of 8th week when bifidobacterium intervention were finished.
- Page 4, in line 137 is written that tissues were homogenized in PBS containing protease and phosphatase inhibitors. However, protein extraction from tissues in the absence of detergents (such as SDS, Triton etc.) is a somewhat unusual procedure.
Response 4: Thank you for your professional suggestion. In this work, ELISA measurement were performed according to the manufacturer’s protocol. Following the “Treatment Method of ELISA Samples” instructions on the official website, it is recommended to homogenize tissues in pre-cooled PBS containing protease and phosphatase inhibitors. Please see the screenshot below or check on https://www.elabscience.com/List-detail-241.html.
Although reagents from different manufacturers may involve different operating procedures, we totally agree with you that the addition of detergent will facilitate the extraction of protein from tissues. In future research, we will fully consider your constructive comments on the basis of following the manufacturer's Protocol. For example, RIPA buffer (containing 1% Triton and 0.1% SDS) which mentioned in previous study[1] may be a better choice. Thanks again for your rigorous consideration.
- Figure 2 (Figure 3 in revised manuscript). In the legend there is only the title. The symbols of statistical significance are not consistent with the results described in the text (see page 5, line 201-202).
Response 5: We appreciate for your valuable comment. More description for Figure 3 has been added in the revised figure legend. Thank you so much for your careful check, and the statistical significance mistake has been corrected in the revised manuscript. Please see the revised parts in line 208-210 and the added figure legend in Line 212-215.
- Figure 3. While BDNF, FNDC5, SYP, PSD-95, and TGF-beta1 have been measured in hippocampal tissue, IL-6 was measured in serum. Why?
Response 6: We totally understand the reviewer’s concern. Interleukin-6 (IL-6) is regarded as a proinflammatory cytokine and the serum IL-6 concentration plays a central role in inflammation with age[3]. Moreover, IL-6 has also been reported to benefit cognition and regulate neurogenesis[4]. Given the limited hippocampal tissue and the fact that IL-6 can cross the blood brain barrier[5], we only measured the serum IL-6 in this work. We totally agree with your professional suggestion and will measure the concentration of IL-6 both in serum and the brain in our future study according to your advice. Thanks again for your valuable comment.
- Although the extent of the differences found between treatments is not specified for any experiment, nor is the meaning of the symbols used for statistics, in many cases the effects (albeit statistically significant) are very small. This limits the impact of the results and the validity of the conclusions drawn.
Response 7: Thanks for your professional comment. We have added the description of significance in Figures and Figure legends. Please see the revised legends of Figure 2-7. We agree with you that in many cases the effects are modest, albeit statistically significant. However, it is equally clear that emerging RCT trials demonstrated that probiotic administration improves cognitive function in AD and mild cognitive impairment (MCI) patients[5,6]. Probiotics are edible and no side effects, which make them safe enough to be a promising daily intervention approach. Since no therapeutic approach has been proven to completely halt AD progression, the aim of our study is to find an effective dietary intervention strategy to prevent or alleviate cognitive decline at the early stage of AD.
Reference:
- Choi, S.H.; Bylykbashi, E.; Chatila, Z.K.; Lee, S.W.; Pulli, B.; Clemenson, G.D.; Kim, E.; Rompala, A.; Oram, M.K.; Asselin, C., et al. Combined adult neurogenesis and BDNF mimic exercise effects on cognition in an Alzheimer's mouse model. Science 2018, 361, doi:10.1126/science.aan8821.
- Rasmussen P , Vedel J C , Olesen J , et al. In humans IL-6 is released from the brain during and after exercise and paralleled by enhanced IL-6 mRNA expression in the hippocampus of mice[J]. Acta Physiologica, 2015, 201(4):475-482.
- Baier, P.C.; May, U.; Scheller, J.; Rose-John, S.; Schiffelholz, T. Impaired hippocampus-dependent and -independent learning in IL-6 deficient mice. Behavioural Brain Research 2009, 200, 192-196.
- Banks, W.A., Kastin, A.J. & Broadwell, R.D. 1995. Passage of cytokines across the blood-brain barrier. Neuroimmunomodulation 2, 241–248
5.Tamtaji, O.R.; Heidari-Soureshjani, R.; Mirhosseini, N.; Kouchaki, E.; Bahmani, F.; Aghadavod, E.; Tajabadi-Ebrahimi, M.; Asemi, Z. Probiotic and selenium co-supplementation, and the effects on clinical, metabolic and genetic status in Alzheimer's disease: A randomized, double-blind, controlled trial. Clinical nutrition (Edinburgh, Scotland) 2018, 10.1016/j.clnu.2018.11.034, doi:10.1016/j.clnu.2018.11.034.
- Sarkar A , Lehto S M , Harty S , et al. Psychobiotics and the Manipulation of Bacteria–Gut–Brain Signals[J]. Trends in Neurosciences, 2016:763-781.

Round 2
Reviewer 2 Report
The manuscript has been improved. I only recommend an additional language revision.
Author Response
Response to reviewers’ comments
(Manuscript ID: nutrients-1179570, Title: Administration of Bifidobacterium breve improves the brain function of Aβ1-42-treated mice via modulation of the gut microbiome)
Dear Editors and Reviewers,
Thank you for your professional suggestions, which are very helpful for revising and improving our paper. In order to make our article more readable, English language editing for the manuscript has been done by MDPI. We have studied comments carefully and revised this manuscript accordingly. Response to your specific comment is detailed below.
Sincerely yours,
Gang Wang
School of Food Science and Technology
Jiangnan University
Wuxi, Jiangsu, 214122, P.R. China
Tel: (86)510-85912155
Fax: (86)510-85912155
E-mail: wanggang@jiangnan.edu.cn
Comments to the Author:
The manuscript has been improved. I only recommend an additional language revision.
Response: We regret there were problems with the English. According to your professional comment, we have carefully revised the manuscript. Moreover, to improve the grammar and readability, we have used the English language editing services of MDPI and received an English editing certificate (ID 29627). Since MDPI uses experienced and native English speaking editors, we hope our carefully revised and edited article will suitable for publishing in a scholarly journal. Attached below is the English editing certificate. Thanks again for your valuable comment. We look forward to hearing from you.
